# Study of Pathogenesis Using Fluorescent Strain of *Cordyceps farinosa* Revealed Infection of *Thitarodes armoricanus* Larvae via Digestive Tract

**DOI:** 10.3390/insects13111039

**Published:** 2022-11-10

**Authors:** Chaoqun Tong, Junhong Wei, Guoqing Pan, Chunfeng Li, Zeyang Zhou

**Affiliations:** 1State Key Laboratory of Silkworm Genome Biology, Southwest University, Chongqing 400715, China; 2Chongqing Key Laboratory of Microsporidia Infection and Prevention, Southwest University, Chongqing 400715, China; 3College of Life Sciences, Chongqing Normal University, Chongqing 401331, China

**Keywords:** *Cordyceps farinosa*, genetic transformation, *Thitarodes armoricanus*, invasion approach, pathogenesis, entomopathogenic fungi

## Abstract

**Simple Summary:**

*Cordyceps farinosa* is a widely distributed entomopathogenic fungus, which is highly infective and lethal to *Thitarodes armoricanus* larvae, thus it seriously endangers the Chinese cordyceps industry. Due to the lack of understanding of its infection process, there are currently no effective prevention and control strategies. In this study, we firstly obtained fluorescently labeled *C. farinosa* mutants by *Agrobacterium*-mediated transformation to better observe the pathway of *C. farinosa* infecting the *T. armoricanus* larvae. We then used the fluorescent *C. farinose* to infect *T. armoricanus* larvae. The results showed that the infection routes were through the body surface and the digestive tract. Our findings lay the foundation for further research and control of this pathogenic fungus.

**Abstract:**

*Cordyceps farinosa* is often utilized as a biocontrol agent because of its wide host range, strong lethality, and safety for mammals. Artificial rearing of *Thitarodes armoricanus* larvae is a prerequisite for the artificial cultivation of *Chinese cordyceps*, and *C. farinosa* is the most lethal pathogenic fungus during the rearing process. However, the infection process of *C. farinosa* is still unclear. In this study, we cloned the promoter of the *C. farinosa* glyceraldehyde 3-phosphate dehydrogenase gene, constructed the EGFP expression cassette, and integrated it into the *C. farinosa* genome via *Agrobacterium* transformation. We obtained a fluorescent strain for better observation of the infection process. Using two different inoculation methods of the fluorescent strain, we observed the traditional infection process through the body surface as well as through the digestive tract via feeding. Both infection modes can lead to larval death and mummification. Our findings demonstrated that during the artificial rearing of *T. armoricanus*, preventing *C. farinosa* pollution should be an important part of the disinfection of the rearing environment.

## 1. Introduction

Chinese cordyceps (also called “Dong Chong Xia Cao”) is used as a traditional Chinese medicine. It is a fungus–insect complex formed by the entomopathogenic fungus *Ophiocordyceps sinensis* (Berk.)-parasitizing soil-borne larvae of *Thitarodes armoricanus* Oberthür. In recent years, its application range and efficacy have received increasing attention [1,2]. Owing to the scarcity of wild Chinese cordyceps, artificial cultivation has become the remedy to this [3,4]. The artificial rearing of *T. armoricanus*, the host insect of *O. sinensis*, is a key requirement for the artificial cultivation of Chinese cordyceps [5]. However, the artificial rearing of *T. armoricanus* larvae often suffers from the harm caused by several pathogens during the feeding process, with *Cordyceps farinosa* being the most harmful pathogen [6].

*Cordyceps farinosa* (homotypic synonym: *Isaria farinosa*) is a pathogen of *T. armoricanus* larvae with a mortality rate up to 90%. It can infect insects at all stages, yet no effective control measures have been reported [7]. Pathogenic fungi are believed to infect host insects through two major ways. One is penetration; the pathogen can penetrate the body surface via spore adhesion, germination, and the production of secretory enzymes upon contacting the host cuticle [8]. The other way is entering the digestive tract with food at the time of feeding and consequently penetrating the midgut to complete the infection [9]. There are more studies on the penetration and body surface contact infection route. It is generally accepted that the conidia of fungi adhere to the insect’s outer integument by hydrophobic forces [10]. After the conidia are adsorbed, an appressorium structure is created under suitable conditions, which makes them more tightly adherent to the insect’s outer epidermis [11]. When the temperature and humidity are suitable, spores swell, germinate, and produce germ tubes with water absorption [12]. In the meantime, the spores can secrete various hydrolytic enzymes, such as chitinases, proteases, and esterases [13,14]. As a result, the fungus finally penetrates the host body surface and enters the hemocoel under the action of enzymes and germ tube expansion pressure [15]. After the fungus invades the blood cavity, it proliferates as a hyphal body or hyphae, destroys the tissue cells, and damages the insect’s muscles, fat body, trachea, and other organs [16,17]; Some fungi also synthesize and secrete toxins during the growth process. These toxins affect the host’s physiological and metabolic functions, ultimately leading to the death of the host [18,19]. On the other hand, studies addressing infestation via the digestive tract are relatively scarce. It was reported that *Beauveria bassiana* and *Metarhizium brunneum* can infect insect hosts through the digestive tract via conidia and blastospores, respectively [9,20].

During the artificial rearing of *T. armoricanus* larvae, the larvae die and mummify after infection with *C. farinosa*. *C. farinosa* was confirmed to infect *T. armoricanus* larvae via the spiracles and epidermis by histological observation of its infection process [21]. The pathogen penetrates the larval body surface, forms coremium, and produces many conidia that are ejected into the air [6,22]. However, whether *C. farinosa* can infect the host via the digestive tract remains unknown. The process of the pathogen infecting the host could be easily observed by using transformants expressing fluorescent proteins [23,24,25]. Therefore, in this study, we constructed a fluorescently-labeled *C. farinosa* mutant by the well-established method of *Agrobacterium*-*tumefaciens*-mediated transformation (ATMT) [26,27]. In this way, we were able to easily observe and comprehensively assess the pathogenesis of *C. farinosa* towards *T. armoricanus* larvae.

## 2. Materials and Methods

### 2.1. Strains and Culture Conditions

*Cordyceps farinosa* was isolated from the *T. armoricanus* rearing environment and cultured at 28 °C on potato dextrose agar (PDA) medium. The DH5α strain of *Escherichia coli* used for plasmid amplification was grown in Luria–Bertani (LB) medium containing 100 μg/mL ampicillin or 50 μg/mL kanamycin. The AGL-1 strain of *Agrobacterium tumefaciens* used for fungal transformation was grown on LB medium. Induction medium (IM; 0.03% (*w*/*v*) MgSO_4_·7H_2_O, 0.03% (*w*/*v*) K_2_HPO_4_, 0.03% (*w*/*v*) NaCl, 5% (*v*/*v*) glycerol, 0.18% (*w*/*v*) glucose, 0.2 mM acetosyringone, and 0.78% (*w*/*v*) 2-(N-morpholino)ethanesulfonicacid, pH 5.3) was used to pre-induce virulence of *A. tumefaciens*. Co-cultivation medium (CoM; IM containing 2% agar) was used to co-cultivate *A. tumefaciens* and *C. farinosa*. CZA medium (0.3% (*w*/*v*) NaNO_3_, 0.05% (*w*/*v*) KCl, 0.05% (*w*/*v*) MgSO_4_·7H_2_O, 0.0001% (*w*/*v*) FeSO_4_·7H_2_O, 0.1% (*w*/*v*) K_2_HPO_3_, 3% (*w*/*v*) sucrose, and 2% (*w*/*v*) agar) was used to screen transformants.

### 2.2. Cloning of the Glyceraldehyde 3-Phosphate Dehydrogenase (Gapdh) Gene and Promoter of C. farinosa

*Cordyceps farinosa* was cultured in PDA medium for 10 days at 28 °C for DNA and RNA extraction. Genomic DNA was isolated using the CTAB method, and total RNA was extracted using a Total RNA Kit II (R6934-02; Omega, Norcross, GA, USA). Then, 2 μg of total RNA from each sample was used for reverse transcription synthesis of cDNA with Hifair III 1st Strand cDNA Synthesis SuperMix for quantitative polymerase chain reaction (qPCR; YEASEN, Shanghai, China).

The *C. farinosa* (GCA_000733625.1) genome was searched using Tblastn (https://blast.ncbi.nlm.nih.gov/Blast.cgi (accessed on 1 November 2022)) with the *B. bassiana gapdh* gene (AAT80324) as a protein query. Then, sequence 1.5 kb upstream of the initiation codon (ATG) was cloned as the promoter sequence. The PCR product was purified and cloned into a pESI-Blunt (YEASEN, Shanghai, China) for sequencing (Sangon Biotech, Shanghai, China). The primers used for PCR amplification are listed in Appendix A.

### 2.3. Construction of Binary Vectors

To construct the pCam-BE binary vector, pCambia0380 was used as the backbone. The open reading frame of enhanced green fluorescent protein (EGFP) was controlled by the *gapdh* promoter from *C. farinosa*. The open reading frames of the selectable marker phosphinothricin resistance gene (*bar*) were controlled by the anthranilate synthase multifunctional protein (trpC) promoter from *Aspergillus nidulans*. The primers used for PCR amplification and enzymes used for vector construction are listed in Appendix A.

### 2.4. Agrobacterium-Mediated Transformation of C. farinosa 

The transformation procedure was adapted from the transformation of *B. bassiana* [28]. Briefly, spores grown on PDA medium for 10 days in the dark at 28 °C were collected as transformation materials. To find the appropriate concentration for screening transformants, spores were cultured at different concentrations of glufosinate ammonium (PPT, Bayer, Germany), and 1.15 mg/mL PPT was determined as the proper concentration. *A. tumefaciens* strain AGL-1 harboring pCam-BE vector was cultured at 28 °C on a rotatory shaker (120 rpm) in 50 mL of LB medium supplemented with 50 μg/mL rifampin and 50 μg/mL kanamycin to an optical density of 0.5 (OD_600_). *A. tumefaciens* cells were harvested via centrifugation at 8000× *g*, washed once with IM, resuspended in IM to an OD_600_ of 0.15, and then grown at 28 °C on a rotatory shaker (120 rpm) for 6–8 h. *A. tumefaciens* and *C. farinosa* spore suspensions (10^5^ spores/mL) were co-cultured at a ratio of 1:1. The co-cultures were spread on nitrocellulose membranes on top of CoM plates and incubated for 72 h at 28 °C in the dark. Following co-culture, the membranes were transferred to CZA plates supplemented with 200 μg/mL cefotaxime and 1.15 mg/mL PPT to select positive *C. farinosa* transformants.

### 2.5. Genetic Stability and Fluorescence Analysis of the Transformants

To determine the genetic stability of the transformants, ten transformants were randomly obtained and cultured in PDA medium without PPT. Mycelia were picked from the edge of the colony and transferred to new PDA plates. After repeating this process five times, mycelia from each transformant were transferred to PDA plates containing PPT (1.15 mg/mL). Clones were selected after multi-generation culturation, genomic DNA was extracted using the CTAB method, and the presence of *EGFP* and *Bar* genes was confirmed via PCR using the primers listed in Appendix A. Fluorescence images of mycelial and spore samples were obtained using a fluorescence microscope (Olympus BX53F, Tokyo, Japan).

### 2.6. Inoculation of T. armoricanus Larvae with C. farinosa 

*Cordyceps farinosa* with stable genetic markers was selected and used to prepare spore suspensions at a concentration of 10^8^ conidia/mL with 0.05% Tween-80. The 4th instar larvae (body length around 3 cm) were selected for inoculation with *C. farinosa* via surface immersion and digestive tract feeding. For surface immersion, the larvae were immersed in the spore suspension for 20–30 s, removed, air-dried, and reared on a single head in a petri dish with carrot. For alimentary canal feeding, carrots were cut into small pieces (0.5 cm^3^), dipped in the spore suspension for 1 min, dried, and fed to 4th instar larvae that had been starved for 1 week. Sixty larvae were used per experiment (30 in the experimental group and 30 in the control group). All larvae were reared at 18 °C and 90% humidity. The larvae of the control group were treated with Tween-80 without spores.

### 2.7. Collection and Observation of Infected Larvae Samples 

Twenty-four hours after body surface immersion, the adhesion of *C. farinosa* to the larval epidermis was observed by fluorescence microscopy (Olympus BX53F, Tokyo, Japan) and scanning electron microscopy (Hitachi S-3400N, Tokyo, Japan), and the hemolymph of infected larvae was collected every 24 h for microscopic observation. For digestive-tract-infected larvae, larval excrement was collected two days after inoculation. The samples were resuspended in sterile water and then spread on PDA plate medium (containing 1.15 mg/mL PPT^+^, 50 mg/mL Kan^+^, 100 mg/mL Amp^+^) to detect the colonization of *C. farinosa* in the digestive tract. Then, every two days, larval digestive juice (pressed gently) was checked to detect spore germination. When the larvae were on the verge of death and collecting digestive juice became challenging, the larvae blood samples were collected instead. After the larvae died, the larval intestine was dissected for observation. The mortality rate of infected larvae was counted every day, and when the larvae died, culturation was continued at 28 °C, and the mummification rate was counted. The mummified larvae with fungal mycelium growing on the body surface were observed using a fluorescence microscope (Olympus SZX16, Tokyo, Japan).

### 2.8. Statistical Analysis

Kaplan–Meier survival analysis was applied to estimate the survival rate of host insects. The log-rank test was used to compare the univariate survival curves of the two infection methods. Statistical analysis was performed using OriginPro (2022b SR1 9.9.5.171) statistical software. Values of *p* < 0.01 were considered to be highly significantly different.

## 3. Results

### 3.1. pCam-BE Vector Construction

The *gapdh* gene of *C. farinosa* was cloned using gDNA and cDNA as the templates. After homologous alignment and verification (Appendix A), the results showed that the length of the gene and the coding region were 1351 bp and 1017 bp, respectively (Figure 1A). The sequence upstream of the start codon was cloned as the promoter Pgpd (Figure 1B), which was used to construct the transgenic vector. The schematic of the constructed vector pCam-BE is shown in Figure 1C.

### 3.2. Fluorescent Strain of C. farinosa

After five rounds of resistance screening, transformants were randomly selected, and the genome DNA was extracted. The PCR results showed that the resistance gene *Bar* and the fluorescent gene *EGFP* were integrated into the genome (Figure 2A). A green fluorescent signal was detected in the hyphae and spores, indicating that the EGFP gene could be expressed normally in *C. farinosa* (Figure 2B).

### 3.3. C. farinosa Infection through Host Body Surface

Twenty-four hours after the immersion process in *C. farinosa* spore suspension, the *T. armoricanus* larval epidermis was observed via fluorescence microscopy (Figure 3A) and scanning electron microscopy (Figure 3B). The results showed that the *C. farinosa* spores adhered to the host epidermis and bristles. After four days of infection, the larvae were sluggish, and *C. farinosa* hyphae were observed in the hemolymph (Figure 3C). The hosts died after six days of infection. The originally clear yellowish blood turned viscous creamy white, and numerous hyphae were observed. After eight days of infection, the hosts began mummifying, the blood was semisolid, and interdigitating fungal hyphae were observed under a microscopic view.

### 3.4. C. farinosa Infection via Host Digestive Tract 

After the *T. armoricanus* larvae were inoculated via feeding, larvae excrement was collected, diluted with sterile water, and spread on PDA plate medium (containing PPT^+^, Kan^+^, Amp^+^). The samples were cultured at 28 °C for seven days (Figure 4A). Green fluorescence could be observed in the colonies from the infected group (Figure 4B). The fluorescent colonies were randomly picked, and the genome was extracted and used as a PCR template. The presence of fluorescent transformants of *C. farinose* was confirmed by the EGFP primer and the detection primer Tar-1F/R (Figure 4C). The results demonstrated that *C. farinosa* was able to infect the host by the digestive tract. 

Two days after inoculation, the larval digestive juice was prepared by the extrusion method. The juice was smeared on slides and the fluorescent signal of *C. farinosa* spores was detected by microscope (Figure 5A). After four days, the spores began to swell and germinate. After six days of inoculation, *C. farinosa* began to form a mycelium, and the larval viability decreased and became moribund. After seven days, the larvae died, and fungal fluorescence signals were detected in the hemolymph, gut, and trachea (Figure 5B). After nine days, *C. farinosa* proliferated massively, and the larvae began to mummify.

### 3.5. Comparison and Statistical Analysis of the Two Infection Methods 

The dead larvae were placed at 28 °C seven days after they mummified. Both ways of infections could make the hosts become covered with white hyphae on their body surfaces (Figure 6A). These results indicated that both methods could be used to infect *T. armoricanus* larvae with *C. farinosa*. The Kaplan–Meier survival analysis of larval mortality revealed that epidermal infection led to an earlier onset of host death and a higher mortality rate, with highly significant differences in the survival curves (*p* < 0.01) among the two infection methods and the control (Figure 6B). In contrast, digestive infection led to a later onset of host death and a lower mortality rate, but a higher mummification was reached, i.e., 86.67% (Figure 6C). 

## 4. Discussion

*Cordyceps farinosa* is the most harmful pathogenic fungus during the artificial rearing of *T. armoricanus* larvae, but its infection pathway and pathogenic process remain unclear. We have previously tried to use wild-type *C. farinosa* for infection experiments directly. However, it is difficult to determine whether the infection is successful by observing the infection and to assess the proliferation process in real time. Alternatively, the above analysis could be achieved by using fluorescent-labeled transgenic fungi [29,30].

Therefore, in this study, we manipulated the genetic operation system of *C. farinosa* and constructed a fluorescent mutant strain with EGFP. Through two different ways of infection, we not only found that *C. farinosa* could infect the host through epidermal contact but also proved for the first time that *C. farinosa* could infect via the host’s digestive tract.

We demonstrated that *T. armoricanus* larvae were infected by *C. farinosa*, and then killed and mummified by both infection methods. In comparison, the mortality rate of epidermal infection was higher (96.67%) than that of digestive tract infection (90%). However, the larval mummification rate was lower (70%) than that of digestive tract infection (86.67%). We showed that the pathogen penetrated the epidermis and then entered the insect’s hemolymph. We believe that in this way, the host’s immune system was undermined, which thus lead to a higher mortality rate. At the same time, the midgut structure was destroyed with the proliferation of the fungi. When the intestinal microbiota of *T. armoricanus* is dramatically disturbed and dominated by bacteria, insects tend to rot rather than mummify [31,32]. In comparison, the lower mortality rate with digestive tract infection could have resulted from the lower amount of inoculated *C. farinosa* because of the low feeding volume of *T. armoricanus* larvae. After *C. farinosa* enters the host’s digestive tract, it must co-compete with other microorganisms in the host’s gut for a niche, and it may also be inhibited by other microorganisms [33], leading to lower infection and mortality rates [34]. Yet, the establishment process gives the host time for gentle microbiota reformation, and when the dominant group becomes *C. farinosa*, the mummification rate of the larvae would be higher.

It is believed that the conidia of entomopathogenic fungi adhere to the host surface by non-specific hydrophobic forces and secrete enzymes to help adhesion [35]. In addition, some invasive structures, such as appressoria form, which exert mechanical pressure to help penetrate the host and also secrete a combination of enzymes such as chitinases, proteases, and esterases [36,37]. In our study, the adherence of *C. farinosa* spores to the larval epidermis was observed, but the formation of infection structures such as appressoria was not observed. When *C. farinosa* infects the host via the digestive tract, the greatest challenge is to proliferate and form a superior econiche in the host gut environment. As an entomopathogenic fungus, *B. bassiana* secretes oosporin during infestation of the host, which can inhibit bacterial proliferation, thereby allowing *B. bassiana* to become a dominant microbial population [38]. Similarly, *Metarhizium anisopliae* secretes destruxin to suppress host immunity while being lethal to low-instar larvae [39,40]. *C. farinosa* can secrete various lipases, proteases, and chitinases [41,42]. Yet, whether these enzymes or any secreted metabolites play a role in the infection of *C. farinosa* requires further elucidation.

For further applications, *C. farinosa* has greater application potential compared to other entomopathogenic fungi such as *B. Bassiana* because of its higher efficiency in controling agricultural and forestry pests in cold environments [41,43,44], and because of the multiple infection routes we proved in this study. Therefore, establishing the genetic manipulation system, as we did in this study, could enable researchers to have a better understanding of this pathogen and enable its better utilization in pest biocontrol. In addition, the mechanism of *O. sinensis* infection of *T. armoricanus* and the consequent mummification process is also unknown [45,46]. Researchers believe that the infection routes of *O. sinensis* and *C. farinose* to *T. armoricanus* are similar [47,48]. Considering the extremely slow growth of *O. sinensis*, *C. farinosa* can be used as a reference model to provide insights on *O. sinensis* studies in the near future.

## 5. Conclusions

In this study, we constructed an *EGFP* expression vector, explored the genetic operation system based on the *Agrobacterium-tumefaciens*-mediated transformation method in *C. farinosa*, and successfully constructed fluorescent-labeled transgenic *C. farinosa*. The infection experiment of the fluorescent strain proved that *C. farinosa* could infect the *T. armoricanus* larvae via both the digestive tract and epidermis. In comparison, the mummification rate of infection via the digestive tract was higher than that of infection via the body surface. The mortality rate of infection via the body surface was higher than that of infection via the digestive tract. This study provides insights for further investigations of the prevention and control of this fungi.

## Figures and Tables

**Figure 1 insects-13-01039-f001:**
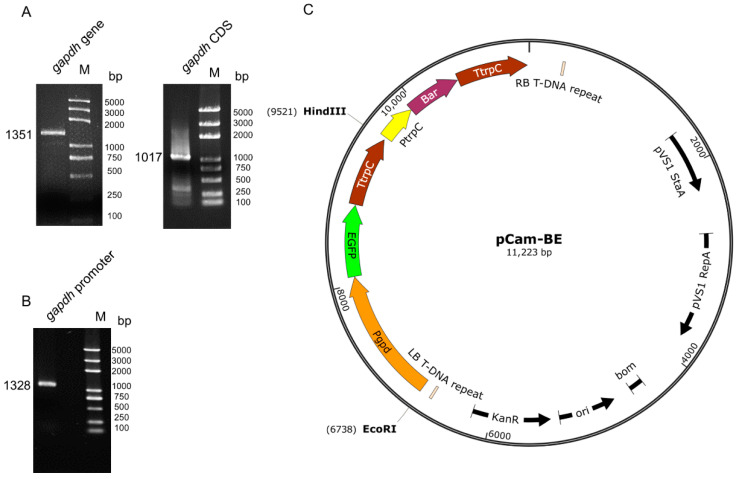
Promoter cloning and vector construction. (**A**) Cloning results of *C. farinosa* glyceraldehyde 3-phosphate dehydrogenase (*gapdh*) gene, using genomic DNA (gDNA) and complementary DNA (cDNA) as templates to clone the full-length and coding sequence (CDS) of the gene, respectively. (**B**) Cloning results of the *gapdh* gene promoter region of *C. farinosa*. (**C**) Schematic of constructed vector pCam-BE. The vector contained a selectable gene (*Bar*) and reporter gene (enhanced green fluorescent protein, *EGFP*). The *Bar* gene was under the control of the *Aspergillus nidulans* anthranilate synthase multifunctional protein (*trpC*) promoter, and the *EGFP* gene was under the control of the *gapdh* promoter. The restriction enzyme sites used are also shown in the figure.

**Figure 2 insects-13-01039-f002:**
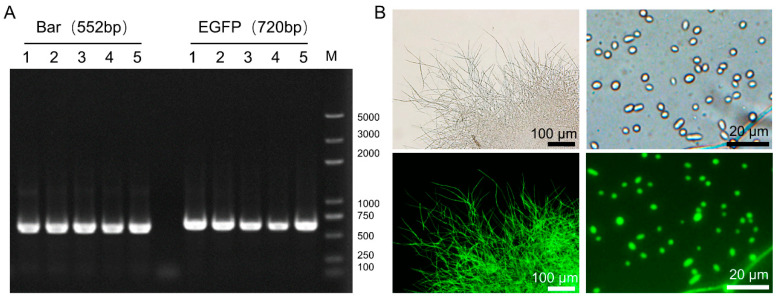
Verification of *C. farinosa* transformants. (**A**) Amplification of total DNA extracted from putative transformants with *Bar* and *EGFP* primers. (**B**) Fluorescence observation of putative transformants.

**Figure 3 insects-13-01039-f003:**
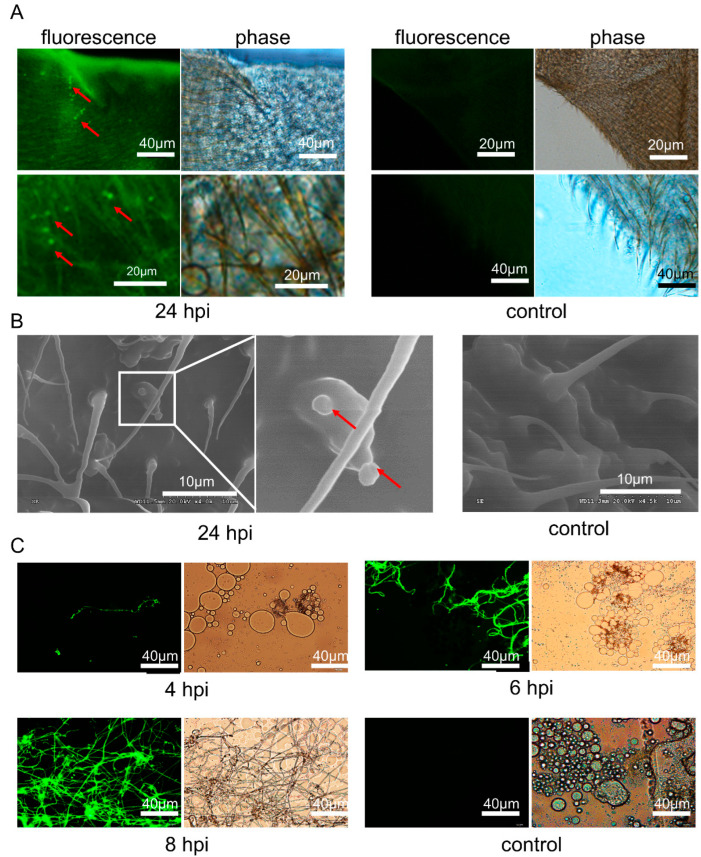
Invasion process of *C. farinosa* via the body surface. (**A**) Fluorescent microscopic observation of larval epidermis after 24 h of body surface infection; red arrows indicate the *C. farinosa* spores. (**B**) Scanning electron microscopy of the epidermis of larvae 24 h post-infection via the body surface; red arrows indicate the *C. farinosa* spores adhering to the epidermis and bristles. (**C**) Observation of host hemolymph at different time points after body surface infection; green fluorescent signals indicate the transgenic *C. farinosa*. dpi: days post-infection; hpi: hour post-infection; control: untreated larvae.

**Figure 4 insects-13-01039-f004:**
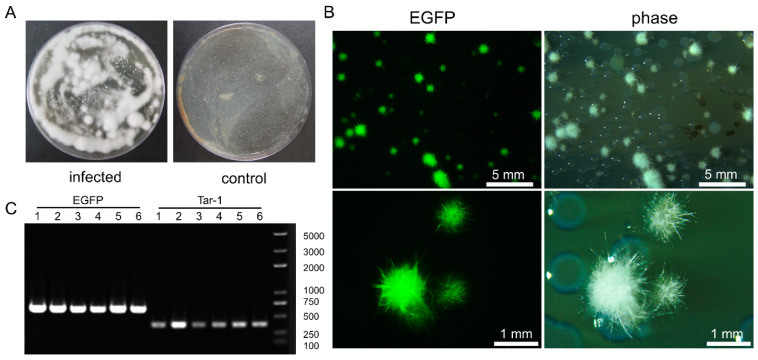
*C. farinosa* detection in the excrement of *T. armoricanus* larvae. (**A**) Microbial cultures of the excrement of infected and control larvae in potato dextrose agar (PDA) medium (containing PPT^+^, Kan^+^, Amp^+^) at 28 °C for seven days. The excrement of uninfected larvae was used as control. (**B**) Fluorescence and white light microscopic observation of colonies on plating medium. (**C**) Fluorescent colonies were amplified by polymerase chain reaction (PCR) using EGFP and Tar-1 primers.

**Figure 5 insects-13-01039-f005:**
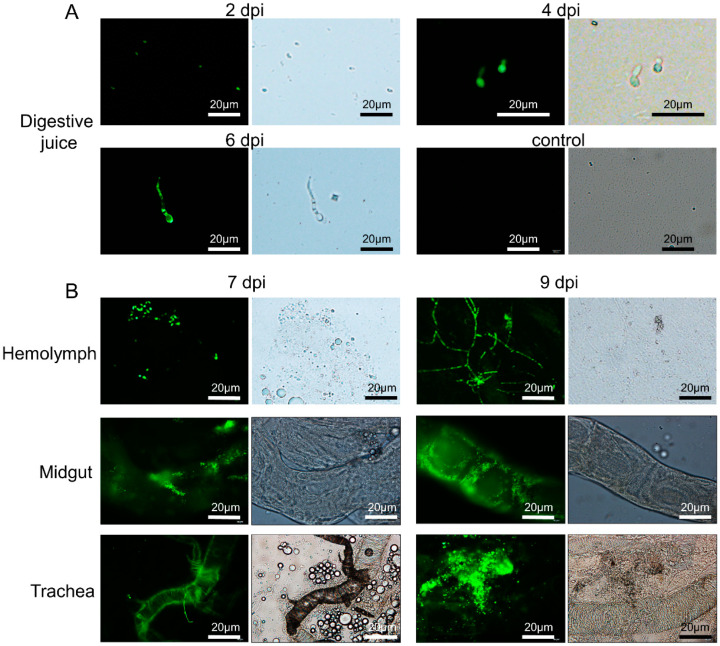
Invasion process of *C. farinosa* via the digestive tract. (**A**) Observation of larval digestive juice at different time points after infection. Uninfected larvae as control (**B**) Observation of larval tissues at different time points after infection. The green fluorescent signal represents the transgenic *C. farinosa*. dpi: days post-infection.

**Figure 6 insects-13-01039-f006:**
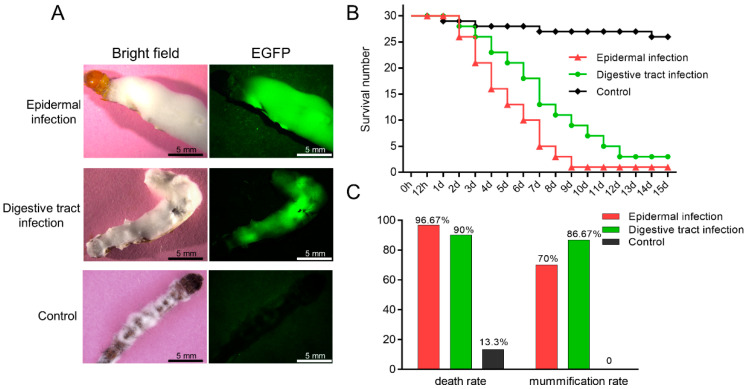
Comparison and statistical analysis of the two infection methods. (**A**) *C. farinosa* caused larval death and mummification via different modes of infection, and white *C. farinosa* hyphae grew on the body surface. Wild type *C. farinosa* was used as control. (**B**) Kaplan–Meier survival curves of *T. armoricanus* larvae with different infection methods. (**C**) Mortality and mummification rates of *T. armoricanus* larvae using different infection methods. Uninfected larvae were used as control.

## Data Availability

All data is provided in the manuscript.

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
