# Peer review of "Study of Pathogenesis Using Fluorescent Strain of *Cordyceps farinosa* Revealed Infection of *Thitarodes armoricanus* Larvae via Digestive Tract"

_insects, 2022, doi:10.3390/insects13111039_

Round 1
Reviewer 1 Report
The authors have carried out an interesting study whose main objective is to study the different pathways of infection of the entomopathogen and compare their lethal effects. In this sense, my major concern is that the Materials and Methods section does not mention the statistical test used to compare the results. It is necessary to mention the analysis performed and give statistical support to the results before considering the manuscript for publication. I am not a native speaker, but I think the grammar of the manuscript should be checked. Below is a list of some of the editorial changes that authors should make:
The current name of Isaria farinosa is Cordyceps farinosa (Holmsk.) Kepler, B. Shrestha & Spatafora Fungus 8 (2): 346 (2017). The authors should modify the name of the species in the title and throughout the entire manuscript.
Line 14: Agrobacterium in Italic.
Line 37: Replace Cordyceps sinensis by C. sinensis.
Line 38: Add “of” before T. armoricanus.
Line 70: Agrobacterium tumefaciens in Italic.
Line 206: Replace spors by spores or conidia.
Lines 212-218: I suspect that the grammatical structure of this paragraph is not correct. Please review it.
Regards
Reviewer 2 Report
The manuscripts presents results of pathogenesis of entomopathogenic fungus Isaria farinosa which cause serious problems in mass rearing of T. armoricanus larvae used for production of traditional Chinese medicinal fungus. The authors did successful transformation of I. farinosa to obtain fluorescent strain. Using this strain pathogenesis was observed after two ways of infection - via surface treatment and via treatment through digestion. Although infection through insect cuticle is well known in entomopathogenic fungi, the other pathways have been less elucidated so the results are novel and worth to be published.The manuscript is well written, methods are sound (statistical analysis is missing), bioassays performed in reasonable number of replications and results are documented by numerous figures (some of which are too small/of low quality) and supplementary material. However, revision is needed before this submission can be accepted for publication - see specific comments below. The manuscript fits well into the scope of Insects but it could also fit well into Journal of Fungi (JoF).
Specific comments (numbers indicated line of MS):
2-3 Title: Current fungus species name is Cordyceps farinosa (see mycobank.org) so I suggest you replace it through whole manuscript and mention synonymum Isaria farinosa along with first occurrence of the name in Introduction.
Title should be also modified to reflect that fluorescent strain was obtained in the study and used in bioassays, e.g. "Study of pathogenesis using fluorescent strain of Cordyceps farinosa revealed infection of Thitarodes armoricanus larvae via digestive tract".
10 entomogenous > entomopathogenic
22 I. farinose > I. farinosa
29 is any new idea mentioned in ms? If yes, it could be indicated or the sentences needs modification.
30 other keywords worth to be included: "pathogenesis" (it not written in title as suggested above), "entomopathogenic fungi"
33 Current name is Ophiocordyceps sinensis (Berk.) G.H. Sung, J.M. Sung, Hywel-Jones & Spatafora (see mycobank.org), I also suggest to write name(s) of author(s) who described the species when the species name is mentioned for the first time. and subsequently use abbreviated name, i.e. O. sisensis (except when it is at the beginning of sentence).
36 add author name of T. armoricanus
38 rearing _of_ T. armoricanus
41 I. farinosa > Cordyceps farinosa; lethality > mortality
45 body wall > cuticle
66 Cordyceps farinosa (beginning of sentence)
78 Cordyceps farinosa
78 could you specify how it was isolated, i.e.g. conidia from insect cadavers and how it was identified (microscopic and macroscopic characteristics, genetic analysis?) If genetics was analysed, please provide genbank accession number(s). Where is the original and transformed strain deposited?
93 for 10 days
139 please write microscope model and city and country of manufacturer
142 > to prepare spore suspensions at concentration 10^8 conidia/ml ...
165 mycosis on cadavers was not observed?
Figs. 1-6: most photos are too small for reading of labels, please increase their size and resolution
209-210 controls: untreated larvae. (delete last sentence as it is redundant).
211 > using the digestive tract infection-method
213 > ... water, was spread ...
215 > selected fluorescent
239 > seven days after which they mummified and were covered ...
241-248 in the fact, there is no statistics, you need to use some test, e.g. Kaplan-Meier survival analysis model, estimate LT50, compare mortality by chi-square test and specify the test and software used also at the end of Material and Methods
250 as written above, proper statistical analysis needs to be conducted.
262 there are other examples when fluorescent strain of entomopathogenic fungus (e.g. M. anisopliae) was used for study of infection/pathogenesis so some might be cited, too.
290 > entomopathogenic
Round 2
Reviewer 1 Report
No comments
Reviewer 2 Report
The authors addressed all my suggestions, added statistical analysis of survival data, added missing information and overall improved the quality of the manuscript. I advice to accept revised version for publication in Insects.